ecology

plant–herbivore interaction, mesograzers, kelp, Ampithoidae, Phaeophyceae, algal detritus

**Author for correspondence:**
Lars Gutow
e-mail: lars.gutow@awi.de

# Small burrowing amphipods cause major damage in a large kelp

Lars Gutow[1], Alistair G. B. Poore[2], Manuel A. Díaz Poblete[3], Vieia Villalobos[3] and Martin Thiel[3,4,5]

[1]Department of Functional Ecology, Alfred Wegener Institute Helmholtz Centre for Polar and Marine Research, 27570 Bremerhaven, Germany
[2]Evolution & Ecology Research Centre, School of Biological, Earth and Environmental Sciences, University of New South Wales, Sydney, NSW 2052, Australia
[3]Facultad de Ciencias del Mar, Universidad Católica del Norte, Coquimbo, Chile
[4]Millennium Nucleus of Ecology and Sustainable Management of Oceanic Islands (ESMOI), Coquimbo, Chile
[5]Centro de Estudios Avanzados en Zonas Áridas (CEAZA), Coquimbo, Chile

LG, 0000-0002-9017-0083

Large herbivores such as sea urchins and fish consume a high proportion of benthic primary production and frequently control the biomass of marine macrophytes. By contrast, small mesograzers, including gastropods and peracarid crustaceans, are abundant on seaweeds but have low per capita feeding rates and their impacts on marine macrophytes are difficult to predict. To quantify how mesograzers can affect macrophytes, we examined feeding damage by the herbivorous amphipods *Sunamphitoe lessoniophila* and *Bircenna* sp., which construct burrows in the stipes of subtidal individuals of the kelp *Lessonia berteroana* in northern-central Chile, southeast Pacific. Infested stipes showed a characteristic sequence of progressive tissue degeneration. The composition of the amphipod assemblages inside the burrows varied between the different stages of infestation of the burrows. Aggregations of grazers within burrows and microhabitat preference of the amphipods result in localized feeding, leading to stipe breakage and loss of substantial algal biomass. The estimated loss of biomass of single stipes varied between 1 and 77%. For the local kelp population, the amphipods caused an estimated loss of biomass of 24–44%. Consequently, small herbivores can cause considerable damage to large kelp species if their feeding activity is concentrated on structurally valuable algal tissue.

## 1. Introduction

A significant proportion of primary production by marine macrophytes is consumed by herbivores [1,2]. Large herbivores such as fish and sea urchins can remove substantial amounts of biomass, with strong consequences for the structure of macrophyte assemblages [3,4]. Marine macrophytes also host diverse assemblages of small herbivores (including peracarid crustaceans and gastropods), collectively known as mesograzers (*sensu* [5]), which use seaweeds as food and habitat [6–8]. Mesograzers often occur in high abundances on macroalgae and seagrasses [9], but their effects on the fitness of individual macrophytes and on the fate of benthic primary production remain difficult to predict.

In many systems, the abundance of mesograzers is strongly controlled by predators, such as fish and shrimps [10,11]. Accordingly, at natural densities, mesograzers are unable to limit the growth of large seaweed species [12]. Generally, the impacts of mesograzers on macrophytes are only evident when released from predator control. As a consequence of exceptional environmental conditions, or in mesocosm experiments, mesograzers can substantially increase in abundance and have strong impacts on the performance and biomass of large macrophytes [13,14]. Additionally, there is growing evidence that small herbivores can have considerable effects on individual macrophytes even at low

abundances if their feeding activity is concentrated on valuable tissues. For example, consumption of photosynthetically active tissue by isopods affects kelp growth [15], whereas excavation of stipes and holdfasts by boring mesograzers can compromise structurally important tissues and provoke substantial biomass losses [16,17].

The impacts of small herbivores are dependent on the distribution of the grazers and their feeding activity [18,19]. Aggregation of herbivores concentrates the feeding activity of consumers in certain areas within a macrophyte bed, or on specific parts of an individual [15]. Fragmentation of the landscape with scattered patches of suitable habitat can result in a clumped distribution of small herbivores with low potential for dispersal [20]. Avoidance of predation [21] or specific reproductive behaviour (e.g. extended parental care) may further promote the aggregation of individuals [22].

Aggregation of conspecific grazers may be particularly intense in species with pronounced microhabitat specialization. Among the mesograzers consuming large macroalgae, many records of grazer damage come from amphipod species in the family Ampithoidae [23]. These amphipods construct burrows in kelp stipes [16] or form nest-like tubes by folding algal blades [24]. The amphipods reside within these domiciles for extended periods of time, feeding on algal tissue [25]. The effects of burrowing and nest-building amphipods on their algal hosts vary substantially among the studied species, from minor damage [26] to local mass mortality of kelp [16]. However, quantitative estimates of the damage induced by amphipods at the level of individuals and local populations of large kelp species are scarce (e.g. [11,27]).

While we would expect that long-term associations between small herbivores and their algal hosts would accelerate the deterioration of plant tissue inside their domiciles [28], extended occupancy may provoke specific responses by the alga [29]. Accordingly, predicting damage by mesograzers on large kelps requires an understanding of the sequence of domicile development following infestation of the host alga, the structure of the grazer assemblages inside the domiciles, as well as the algal responses to damage.

To quantify how burrowing mesograzers can affect a large kelp, we studied herbivorous amphipods on subtidal individuals of the kelp *Lessonia berteroana* in a kelp forest in northern-central Chile (southeast Pacific). Excavation of structurally important parts of the kelp stipes by two co-occurring species of burrowing amphipods suggests that grazing damage may induce further loss of biomass to the kelp. To test this, we first quantified the levels of infestation of *L. berteroana* by amphipods within the kelp forest and within kelp individuals. Second, we described the amphipod assemblages inside the domiciles and the morphological development of the infested thallus parts to understand the responses of the kelp to the herbivores. Third, we estimated the loss of frond biomass induced by amphipod grazing to quantify the damage to individual kelps and loss of kelp biomass across the local kelp population.

## 2. Material and methods

### (a) Structure and development of domiciles and amphipod assemblages

To describe the interaction between stipe-boring amphipods and the kelp *L. berteroana*, we analysed the morphology of the amphipod domiciles and the composition of amphipod assemblages inside the domiciles. Stipes of subtidal fronds of *L. berteroana* were collected in February 2011 and 2014 at Playa Blanca, northern-central Chile (28°10′S, 71°10′W). In total, 81 stipes (29 in 2011 and 52 in 2014) with amphipod domiciles were collected. The stipe sections with the domiciles were identified by a snorkeling investigator, cut off with a knife above and below the adjacent branchings of the stipe, transferred individually into plastic bags and transported in a cooler to the Universidad Católica del Norte (UCN) in Coquimbo. In the laboratory, the stipe sections were preserved in 5% formalin or in 95% ethanol until further examination.

Each stipe section was photographed to document the external morphology of the domicile, which was then used to distinguish four stages of infestation; these stages probably display a temporal sequence following initial infestation (electronic supplementary material, figure S1). In Stage 1, the domicile was apparent by a small hole formed by the initial entrance to the burrow. In Stage 2, the area of the domicile was swollen, occasionally with conspicuous deformations of the infested stipe section. The opening of the domicile was more irregular and expanded. Stage 3 was an advanced stage of tissue disintegration with gaping stretches of the stipe. In Stage 4, further disintegration of the tissue had resulted in breakage of the stipe and loss of the distal stipe sections.

The stipe sections were cut longitudinally to open the domiciles. Some domiciles were conglomerates consisting of several burrows. The number of burrows in each domicile was counted and contrasted between sampling years and among stages of infestation with a generalized linear model (GLM) and negative binomial error distribution. The GLM was run with the manyglm function in the R package mvabund [30], with statistical inference from parametric bootstrapping. Assumptions of the model were checked with plots of residuals versus estimated values.

All amphipods were washed out of the domicile with freshwater and preserved in 5% formalin. The external surface of the stipe and the interior of the plastic bag were inspected for amphipods, which might have abandoned the domicile during sampling and/or preservation. Two amphipod species were identified from within the domiciles: *Sunamphitoe lessoniophila* (Ampithoidae) and an unidentified species of the genus *Bircenna* (Eophliantidae). Individuals of *S. lessoniophila* were identified using [31], counted and classified as adult males (by the presence of well-developed second gnathopods), adult females (by the presence of a fully developed marsupium) and juveniles (lacking these features). Adult females were classified as ovigerous if carrying embryos in the marsupium. In total, 652 individuals of *S. lessoniophila* were isolated from the domiciles comprising 58 males, 32 females and 562 juveniles. Individuals of *Bircenna* sp. were counted (total: 517 individuals) without distinguishing sex or life-history stage.

A photo was taken of each individual together with a scale (millimetre paper). Body length (mm) was measured from the images as the length of the curved dorsal line connecting the tip of the rostrum and the base of the telson using the software package Image Pro Plus. The number of individuals of both amphipod species within the domiciles was contrasted between sampling years, among stages of infestation and among amphipod species with a GLM and negative binomial error distribution.

The population structure of assemblages of *S. lessoniophila* inside the domiciles was described for single burrows. In the conglomerates, which consisted of several burrows, the single burrows were often not clearly separated from each other so that an assignment of the amphipods inside the conglomerate to a specific burrow was not possible. Accordingly, burrows from conglomerates were not considered in the analysis of the assemblages to avoid mixing of assemblages that originally might have inhabited separate burrows.

## (b) Distribution of amphipod domiciles

The infestation rate of subtidal *L. berteroana* by burrowing amphipods across the kelp forest and the distribution of amphipod domiciles within the stipes was quantified from 98 stipes collected at Playa Blanca in March 2017. Ten (in one case eight) stipes each were randomly collected from 10 different rocks (total $n = 98$ stipes) separated by tens to hundreds of metres of boulder substrata, giant kelp *Macrocystis pyrifera* or sandy sediment. Care was taken that all stipes were collected from different holdfasts. The stipes were cut off immediately above the holdfast and transferred into mesh bags.

On the shore, the stipes were inspected for the presence of amphipod domiciles. For each stipe, we recorded the total stipe length and the number of domiciles. To test if domicile density varied between the sampling sites within the kelp forest, the average domicile density was compared between the 10 different rocks by a GLM. Stipe length was treated as an offset to account for the higher probability of a longer stipe to become colonized by amphipods than a shorter stipe. The relationship between stipe length and the number of domiciles was analysed by linear regression. For each domicile, we recorded the stage of infestation and contrasted the number of domiciles among these stages with a $\chi^2$ goodness-of-fit test.

The loss of tissue distal to damage depends on the position of domiciles within each frond. To quantify the position of each domicile along the stipe, we recorded the internode level where the domicile was positioned. Internodes are the straight stipe sections that connect the successive branchings of the stipe. These were numbered in sequence, with level 1 being the most basal internode right above the holdfast (electronic supplementary material, figure S2). The regular arrangement of strictly bifurcate branchings and connecting internodes results in a modular morphology of the kelp thallus that allows for clearly assigning each internode to a specific level. Domiciles positioned in a branching of the stipe were assigned to the internode level above.

To test whether the distribution of domiciles among the internode levels is simply a result of a stochastic encounter of the amphipods with the stipes, we contrasted the distribution of the domiciles with the vertical distribution of kelp biomass along the stipes using a $\chi^2$ goodness-of-fit test. The distribution of biomass along the stipes was estimated from the sum of the number of all internodes at each internode level and the average biomass (g wet weight) of internodes at each internode level. These data were obtained from 40 stipes of *L. berteroana* collected at Playa Banca as described in the following section.

## (c) Estimation of damage across the kelp population

Amphipod domiciles induce breakage of the stipe leading to the loss of the stipe sections and all terminal blades distal to the point of breakage. From sampling alone, it is not possible to know when a stipe section has been lost and how much biomass would be lost in terms of absolute biomass. Therefore, we estimated damage in terms of the 'maximal potential loss of biomass', which is the difference between the observed biomass of the damaged stipe and the biomass expected from a reconstructed undamaged stipe. The undamaged stipe was reconstructed from the damaged stipe making use of the modular morphology of *L. berteroana* (electronic supplementary material, figure S2).

Damage was estimated for 40 subtidal stipes selected randomly from various sites within the kelp forest of Playa Blanca in March 2017. The stipes were cut off immediately above the holdfast, placed in mesh bags and transported in coolers to the UCN in Coquimbo. For each stipe, we counted the number of internode levels, the number of internodes at each level, the number of undamaged and broken-off internodes at each level, the number of breakages at each level that were clearly due to

amphipod domiciles, and the number of breakages at each level that could not be unambiguously attributed to domiciles.

From each of 20 randomly selected stipes, we measured the wet weight (precision: 1 mg) of up to five randomly selected blades allowing for the calculation of the average ($\pm$ s.d.) blade biomass from a total of 94 blades, which was 2.212 $\pm$ 1.738 g per blade. The wet weights (precision: 0.1 g) of 415 internodes from 29 stipes were measured to obtain the average biomass of internodes from all internode levels (for the average internode biomass, see electronic supplementary material, table S1). The internodes were selected to obtain representatives from each internode level while within the internode levels the internodes were selected randomly. Blades and internodes were carefully blotted dry with tissue paper before the individual wet weight was determined. Some terminal internodes were not carrying blades for unknown reasons. Therefore, the proportion of terminal internodes, which carried a blade, was calculated from 29 randomly selected stipes. The average ($\pm$ s.d.) proportion was 82.0 $\pm$ 19.9%.

For the reconstruction of undamaged stipes, we made the simplifying assumptions that (i) the loss of stipe sections had no effect on the subsequent growth of the remaining parts of the stipe, and (ii) the lost parts of the stipe would have grown to the maximum internode level of the remaining part of the stipe. Accordingly, the broken-off stipe section was reconstructed up to the most apical internode level of the remaining stipe. The number of internodes at each internode level was reconstructed according to the average specific propagation rate for each level. The propagation rate for internode level $x$ was calculated for each stipe as the number of internodes at level $x + 1$ divided by the number of undamaged internodes at level $x$. Despite strict bifurcations of the stipe, the propagation rate between two levels was always less than 2 (except for levels 1 and 2) because some internodes did not branch but became distal terminations (for the internode propagation rates, see electronic supplementary material, table S1). The number of distal terminations (TI) at level $x$ was calculated as

$$\mathrm{TI}_x = a_x - \left(\frac{a_{x+1}}{2}\right), \tag{2.1}$$

with $a_x$ being the number of undamaged internodes at level $x$ and $a_{x+1}$ the number of internodes at level $x + 1$. The terminal internodes were stocked with blades according to the average proportion of blade-carrying distal terminations. The total biomass of a stipe was calculated as the sum of the average biomasses of all internodes and blades. The observed biomass of the actual stipe was contrasted with the total expected biomass of the reconstructed stipe to obtain the maximal potential loss of biomass, which was expressed as the per cent expected biomass of the reconstructed stipe. The observed biomasses of the damaged stipes and the expected biomasses of the reconstructed stipes were compared by a paired *t*-test after the differences between the paired values were tested for deviation from a Gaussian distribution using a D'Agostino–Pearson omnibus $K^2$ test ($n = 17$, $K^2 = 0.17$, $p = 0.08$). Finally, the maximal potential loss of biomass due to amphipod domiciles was calculated for the entirety of the 40 stipes from Playa Blanca as the percentage difference between the sum of the total observed biomass of all stipes (i.e. the sum of undamaged and damaged stipes) and the sum of the total reconstructed biomass of all stipes (i.e. the sum of undamaged stipes and the damaged stipes with the lost biomass reconstructed).

In addition to the breakages that were clearly due to amphipod burrows, some stipes showed several additional breakages due to unknown reasons. It cannot be excluded that these breakages were originally also induced by amphipod burrows and that subsequent wound healing had masked the former presence of the burrows. To obtain a maximum estimate of the potential loss of biomass due to all stipe breakages (including amphipod burrows), the above calculations were also made for all breakages (D'Agostino–Pearson normality test: $n = 24$, $K^2 = 2.79$, $p = 0.25$).

The loss of biomass due to breakages induced by amphipod domiciles and due to all breakages was visualized for different numbers of breakages using the following sigmoid regression model:

$$f(x) = a + \frac{(b-a)}{1 + \exp((c-x)/d)},$$ (2.2)

with $a$ being the minimum biomass loss of stipes with no breakage, which was set equal to zero. $b$ is the projected maximum loss of biomass, $c$ denotes the halfway loss of biomass between $a$ and $b$ and $d$ is the slope of the curve.

The same sigmoid regression model was used to visualize the observed biomass of stipes, the loss of biomass due to stipe breakage induced by amphipod domiciles and the loss of biomass due to all stipe breakages for stipes of different size (number of internode levels), with $a$ being the minimum biomass of the smallest stipe, which was set equal to zero. $b$ is the maximum biomass of the largest stipe, $c$ denotes the halfway biomass between $a$ and $b$, and $d$ is the slope of the curve.

## 3. Results

### (a) Structure of domiciles

The 81 amphipod domiciles collected in the kelp forest of Playa Blanca comprised 42 single burrows and 39 conglomerates, each consisting of two to eight single burrows. The number of burrows varied significantly between domiciles of different stages of infestation (figure 1a; for the results of the GLM, see electronic supplementary material, table S2). It was lowest in the early stages but increased substantially in domiciles of Stage 3. In the final Stage 4, the number of burrows per domicile decreased again, probably as a consequence of tissue erosion (and burrow loss) in the broken-off stipe sections. The overall number of burrows per domicile was significantly higher in 2011 than in 2014.

### (b) Amphipod assemblages in domiciles

The number of amphipods varied significantly among domiciles of different stages of infestation (figure 1b; for the results of the GLM, see electronic supplementary material, table S3). Only very few amphipods were found in domiciles of Stage 1 except for a single domicile, which was inhabited by 64 *S. lessoniophila*. The number of *S. lessoniophila* increased substantially in domiciles of Stage 2, whereas the number of *Bircenna* sp. remained low. Subsequently, the number of *S. lessoniophila* declined and remained low in the later stages when the stipe tissue disintegrated and finally broke off. By contrast, the number of *Bircenna* sp. increased in domiciles of Stages 3 and 4 such that the amphipod assemblages were numerically dominated by *Bircenna* sp. in these late stages of infestation. The shift from an early dominance of *S. lessoniophila* towards assemblages dominated by *Bircenna* sp. resulted in a significant interaction between amphipod species and stage of infestation (electronic supplementary material, table S3). This development was similar in both sampling years.

Adult females shared their burrow with one to several (apparently up to three) cohorts of juveniles (figure 2). Distinct cohorts of juveniles of different body size probably resulted from successive moultings of the juveniles after hatching and may indicate repeated reproduction of the females inside their burrows. Eleven burrows were occupied by adult females and variable numbers of similar-sized juveniles. These

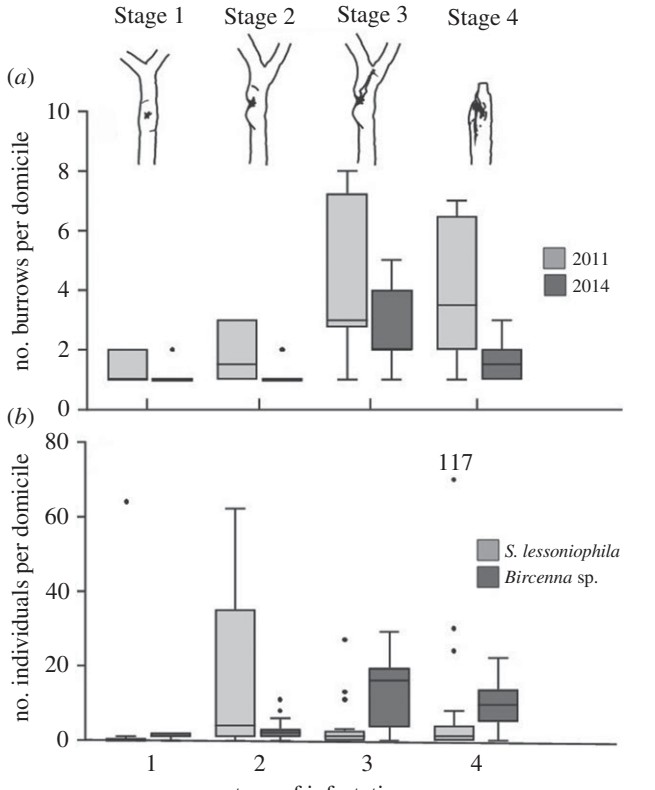

**Figure 1.** (*a*) Number of burrows within amphipod domiciles of different infestation stages (see drawings on top of the figure and electronic supplementary material, figure S1) in stipes of *L. berteroana* from Playa Blanca collected in 2011 and 2014. (*b*) Number of individuals of *S. lessoniophila* and *Bircenna* sp. in domiciles (including single burrows and conglomerates) of different infestation stages in stipes of *L. berteroana*. Data from the sampling years 2011 and 2014 were combined because the pattern was similar in both years. The dot with a number above it represents an outlier at 117 individuals per domicile, which lies outside the scale of the ordinate.

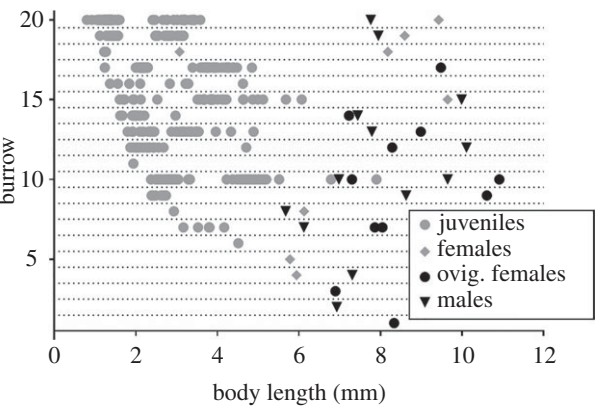

**Figure 2.** Composition of assemblages of *S. lessoniophila* inside individual burrows in stipes of subtidal individuals of *L. berteroana* (*n* = 20). Assemblages from conglomerates were not analysed. Numbering of the burrows is done posterior to sample analysis and only for illustrative purposes.

assemblages of adult and juvenile individuals occurred mostly in burrows of Stage 2, whereas single individuals were also encountered in burrows of Stages 1, 3 and 4.

### (c) Distribution of amphipod domiciles

The number of domiciles per stipe varied significantly among sampling sites in the kelp forest of Playa Blanca

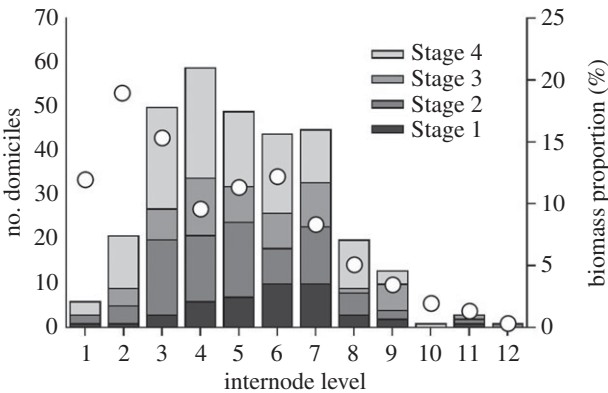

**Figure 3.** Vertical distribution of amphipod domiciles ($n = 312$; columns—left ordinate) on the internode levels along stipes ($n = 98$) of subtidal individuals of *L. berteroana* from Playa Blanca and vertical percentage distribution of *L. berteroana* biomass (circles—right ordinate) within the stipes.

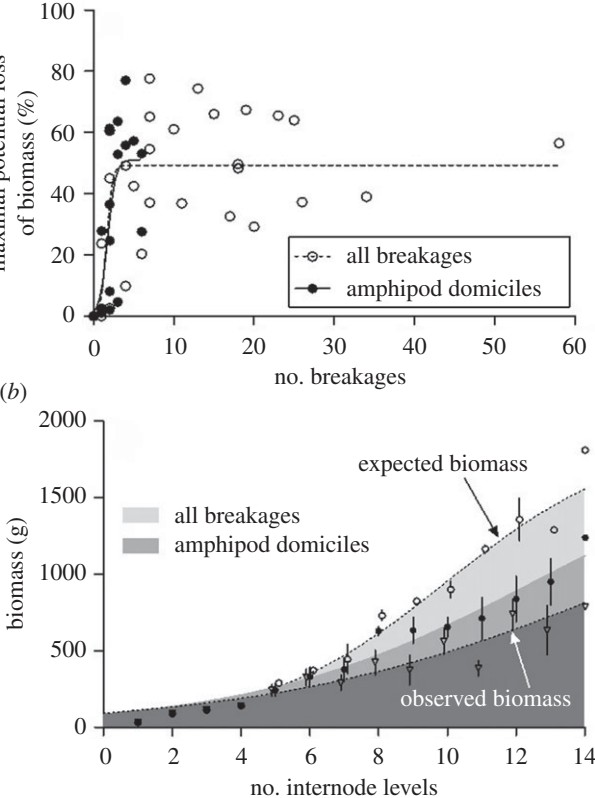

**Figure 4.** (*a*) Estimated maximal potential loss of biomass (%) in stipes of *L. berteroana* due to stipe breakages clearly induced by amphipod domiciles and due to all breakages. (*b*) Observed (triangles) biomass of stipes of *L. berteroana* from Playa Blanca (including undamaged and damaged stipes) and expected (circles) biomass of undamaged stipes (including undamaged stipes and damaged stipes with the lost biomass being reconstructed). The average ($\pm$ s.e.m.) biomass was reconstructed for stipes of different size (no. of internode levels) with damage induced by amphipod domiciles (filled circles) and with damage due to all breakages (including breakages due to amphipod domiciles and breakages due to unknown reasons—open circles). $n = 40$.

(GLM: d.f. = 88, deviance = 87.68, $p < 0.01$; electronic supplementary material, figure S3). A total of 312 domiciles were identified from 98 stipes of *L. berteroana* in 2017, with the proportion of stipes with domiciles varying from 10% to 100% among the sampling sites. The median of the number of domiciles per stipe varied between the sampling sites from 0 (range: 0–1) to 7 (range: 3–13).

About one-third (30.6%) of the stipes were completely free of domiciles. A further 30.6% had one or two domiciles. The maximum number of domiciles per stipe was 23. The positive relationship between stipe length and the number of domiciles was statistically significant, but explained only little of the variance ($F_{1,96} = 16.25$, $p < 0.01$; $R^2 = 0.145$; electronic supplementary material, figure S4).

Within individuals of *L. berteroana*, the majority of the amphipod domiciles were located in the central sections of the stipes (figure 3). Around 79.2% of the domiciles were found at the internode levels 3–7, with few located in the uppermost and in the lowermost stipe sections. The vertical distribution of domiciles differed significantly from the vertical distribution of *L. berteroana* biomass in the kelp forest ($\chi^2 = 58.5$, d.f. = 11, $p < 0.01$). The kelp biomass was primarily located in the large basal and central internodes and declined towards the small apical internodes of the stipes.

The number of domiciles belonging to each stage of infestation differed from an even distribution ($\chi^2 = 23.60$, d.f. = 3, $p < 0.01$). The final stage of infestation (Stage 4) that had already induced stipe breakage and loss of the distal stipe sections was the most frequent stage (40.1% of the domiciles). The second stage where the stipe had started to swell around the entrance of the burrow was the next most abundant (27.2%). Relatively low numbers of domiciles were in the first (14.1%) and in the third (18.6%) stage of infestation, respectively.

The proportion of stipe breakages (i.e. Stage 4 domiciles) was highest in internode levels 3–6, suggesting that amphipods prefer the lower central levels during the initial colonization. Similarly, the proportion of Stage 2 domiciles was highest in internode levels 3–5. The proportion of young Stage 1 domiciles increased steadily from internode level 2 to internode level 7, suggesting that the amphipods progressively move upwards along the stipe upon disintegration of the original domicile in the lower central stipe sections.

## (d) Estimation of damage across the kelp population

Among the 40 stipes of *L. berteroana* collected in 2017 at Playa Blanca, 27 stipes (67.5%) had breakages with 17 of these stipes (42.5%) showing breakages clearly induced by amphipod domiciles. The number of breakages that were induced by amphipod domiciles varied between one and six per stipe. The total number of breakages per stipe varied between one and 58. Stipe breakage induced a considerable maximal potential loss of biomass. The estimated loss of biomass due to amphipod domiciles varied between 1.2% induced by a single breakage (in a stipe with seven internode levels) and 77.0% induced by four breakages (in a stipe with nine internode levels) (median: 36.5%; figure 4*a*). Even the estimated loss of biomass induced by a single breakage varied substantially. The highest estimated loss of biomass induced by a single breakage (in a stipe with nine internode levels) was 27.9%. The estimated biomass differed significantly between damaged and reconstructed stipes ($t_{16} = 5.38$, $p < 0.01$). The estimated loss of biomass of all stipes induced by breakages due to amphipod domiciles was 24.4%.

When considering all breakages, the highest loss of biomass of 77.6% (median: 48.4%) was induced by seven breakages. The variation in loss of biomass induced by breakages was

particularly high at relatively low numbers of breakages. Above about 10 breakages the estimated loss of biomass was consistently high and never below 29.3%. The estimated biomass differed significantly between damaged and reconstructed stipes ($t_{23} = 8.15$, $p < 0.01$). The estimated loss of biomass of all stipes induced by all breakages was 44.2%.

The sigmoid regression model explained 69% of the variation in the estimated loss of biomass for both the breakages induced by amphipod domiciles and for all breakages. About 50% of the maximal potential loss of biomass, as estimated from the sigmoid function, was reached at 1.66 breakages induced by amphipod domiciles, and at 1.77 breakages when all breakages were considered, demonstrating that already a small number of breakages can cause substantial loss of biomass.

When the lost biomass of the damaged stipes was reconstructed, the biomass increased faster with increasing size of the stipe than did the observed biomass of the stipes without reconstructed damage resulting in the highest difference between observed and expected biomass in the largest stipes (figure 4b). The sigmoid regression model explained 66.0% (for the observed biomass) to 93.4% (for the expected biomass) of the variation.

## 4. Discussion

Burrowing by herbivorous amphipods induced characteristic morphological responses and stipe breakage in subtidal individuals of the large kelp *L. berteroana*. The resultant damage to the kelp was variable and depended on the infestation rate, the spatial distribution of the grazers and the size of the algal thallus. Aggregated feeding by the small grazers on structurally important tissues caused considerable loss in kelp biomass. These results demonstrate that mesograzers can have a profound impact on the fate of benthic primary production in coastal ecosystems.

### (a) Structure and development of domiciles and amphipod assemblages

Stipe burrowing by herbivorous amphipods induced morphological change in *L. berteroana*, which comprised local swellings and deformation of the stipe, tissue degradation, stipe cleavage and, ultimately, breakage of the stipe, resulting in the loss of distal stipe sections. The progressive deterioration from a healthy stipe morphology and the increasing degradation of the algal tissue suggest that the Stages 1–4 display a continuous developmental sequence of the domiciles from initial infestation towards final stipe breakage. Domiciles of Stages 1 and 2 were found in healthy stipe sections demonstrating that, unlike the lysianassid amphipod *Orchomenella aahu* on the kelp *Ecklonia radiata* in New Zealand [32], the amphipod species on *L. berteroana* do not require damaged stipe tissue for the initial infestation.

Most domiciles on *L. berteroana* at Playa Blanca were in Stages 2 and 4, indicating that the initial infestation (Stage 1) as well as severe tissue degradation and stipe cleavage (Stage 3) are ephemeral stages and were, therefore, less frequently encountered. Stipe breakage is irreversible and leads to the accumulation of Stage 4 domiciles until wound healing may mask the actual reason for the breakage.

The domiciles remain in Stage 2 for extended periods of time, which permits aggregation of numerous juvenile

*S. lessoniophila* within the same burrow. The distinct groups of similar-sized juveniles probably represent cohorts of juveniles from successive broods. The cohabitation of adult females (occasionally accompanied by adult males) and one to several distinct cohorts of juveniles in the same burrow provides evidence for extended parental care in *S. lessoniophila* [33]. Extended parental care is common among burrowing crustaceans and facilitates juvenile survival [34]. For example, adult *S. stypotrupetes* share their burrows inside the stipes of the kelp *Laminaria setchellii* for several months with juveniles from up to four successive cohorts [16]. In *S. lessoniophila*, the accumulation of conspecifics promotes the local concentration of herbivore grazing activity. Aggregated grazing by numerous small herbivores on structurally and physiologically valuable tissue can affect algal metabolism and growth [15,35] and cause stipe breakage or dislodgement of kelp thalli [36], potentially resulting in the death of individual kelps and degradation of entire kelp beds [16,32].

Stipe failure may be accelerated in *L. berteroana* by the combined feeding of two coexisting herbivore species inside the domiciles. During the early domicile stages, the amphipod assemblages were numerically dominated by *S. lessoniophila*, suggesting that the presence of *S. lessoniophila* inhibits the presence of *Bircenna* sp. inside the shared burrows. When the large assemblages of *S. lessoniophila* left the burrows, the numbers of *Bircenna* sp. increased in older domiciles, which were in advanced stages of tissue degradation, suggesting that previous grazing by *S. lessoniophila* facilitates the accessibility of the algal tissue for *Bircenna* sp. (see also [37]). Continuous grazing by *S. lessoniophila* and, subsequently, by *Bircenna* sp. may result in the coalescence of neighbouring burrows during late stages of infestation, leading to the formation of spacious conglomerates of burrows, which probably weakens the stipe and facilitates stipe breakage.

### (b) Distribution of grazing damage

The effects of herbivory on plant populations and communities depend on the distribution of grazers at different spatial scales [38]. The distribution of a species is shaped by its movement behaviour relative to the structuring of the landscape, i.e. the distribution and availability of habitat [39]. The amphipod domiciles were not evenly distributed among the local kelp aggregations, which are isolated from each other at Playa Blanca by variable stretches of unsuitable habitat. Some sites within the kelp forest were almost completely free of amphipod domiciles whereas at other sites, all thalli were infested. The local aggregation of herbivores results in an uneven distribution of grazing damage within plant populations [15,19] and can have implications for the structure of seaweed assemblages [40].

The amphipod domiciles were predominantly found in the lower central sections of the stipes but were rare in basal and apical stipe sections. Domiciles might be rare in the apical internodes because these sections of the stipe are younger and less likely to have been colonized by amphipods than older parts of the thallus. However, the vertical distribution of the domiciles differed from the average distribution of the stipe biomass of *L. berteroana* indicating that the burrows were not randomly distributed according to the probability of amphipods encountering kelp biomass (otherwise, the older, basal sections would house the highest amphipod densities). Instead, the distribution of the domiciles may result from a

specific microhabitat preference of the amphipods. Small herbivores select microhabitats within their host plants to optimize nutrition and protection from predation [41]. Some small herbivores are unable to forage on tough algal tissues [42], often found in older sections of algal thalli [43]. Accordingly, high tissue toughness may prevent amphipods from burrowing into the old basal stipe sections of *L. berteroana*. Alternatively, epifaunal invertebrates may avoid the basal sections of large kelp thalli to escape intense predation from benthic predators. Demersal fishes prey intensively on benthic invertebrates in kelp forests of northern-central Chile [44]. Accordingly, the nest-building amphipod *S. femorata* predominantly lives on the apical blades of giant kelp *M. pyrifera* where the amphipods suffer less predation than on basal blades [45].

Domiciles on macrophytes provide shelter from predation [46]. Accordingly, herbivorous amphipods have evolved strategies that enhance the persistence of protective domiciles and minimize the necessity for risky dispersal. For example, *S. femorata* constantly advance their domiciles on blades of *M. pyrifera* towards the intercalary growth meristem at the blade base thereby protracting the loss of the protective domicile at the erosive distal tip of the blade [24]. In *S. lessoniophila*, burrowing in the delicate apical internodes probably leads to rapid stipe breakage forcing the amphipods to abandon their domiciles. By avoiding the apical internodes of *L. berteroana*, the burrowing amphipods enhance domicile persistence and minimize predation risk but concentrate the grazing activity on lower stipe sections where stipe breakage causes the loss of extensive parts of the kelp thallus. The vertical positioning of the nest in the lower central stipe sections thus results from a compromise between (i) the apical internodes (which are too thin to support persistent burrows), and (ii) the massive basal internodes, where high tissue toughness may prevent effective excavation of a safe burrow.

## (c) Consequences of grazer damage to kelp

Grazing by herbivores on structurally important tissues can lead to disproportionate loss of plant biomass [18,47] and predicting plant damage from herbivore consumption rates alone can underestimate the effects of small herbivores on large macrophytes [48]. For example, grazer-induced breakage in the brown seaweed *Ascophyllum nodosum* can exceed the amount of stipe tissue consumed by small herbivores by the factors 20–100 [49]. The missing biomass of single damaged stipes of *L. berteroana* varied between 1% and 77%, depending on the number of breakages per stipe and the positioning of the breakages along the stipes. For an average stipe biomass of about 500 g (as estimated from the reconstructed stipes), the median estimated biomass loss per stipe (36.5%) would amount to about 180 g. Adopting the body mass (0.1 g) and the daily consumption rate (about 50% of the body mass) of adult individuals of the much larger *S. femorata* [45], average numbers of 15 amphipods per domicile and 3.2 domiciles per stipe would result in a daily consumption of about 2 g kelp biomass per stipe. At that rate, it would take about three months to consume the biomass, which is lost by grazer-induced breakage of a single stipe.

The preference of the amphipods for constructing domiciles in the lower central stipe sections induced the loss of extensive proportions of the algal thallus already at low numbers of breakages per stipe. For the entire *L. berteroana* population at Playa Blanca, the amphipods caused an estimated loss of biomass of 24–44%, depending on the proportion of stipe breakages that were assigned to amphipod burrowing. This rate is similar to the estimated biomass loss of 28% in the kelp *Laminaria longicruris* in Nova Scotia [40] but lower than the weight loss of 55–78% in the brown alga *Fucus distichus* in New Brunswick, Canada [50], both resulting from grazing by the herbivorous gastropod *Lacuna vincta*.

In *L. berteroana*, the number of amphipod domiciles correlated positively with stipe length. Accordingly, the estimated loss of biomass due to stipe breakage increased with the size of the stipe. Assuming that the stipes grow throughout the entire lifespan of the kelp, more domiciles can accumulate over time on longer stipes. Moreover, longer stipes offer more opportunities to construct persistent burrows in appropriate stipe sections, which can be reached by the amphipods even by short within-stipe movements. Alternatively, young kelp thalli may have higher levels of chemical defense against herbivores than old thalli, and thus be less preferred by the amphipods [32].

## (d) Conclusion

Our results demonstrate how grazing by small herbivores can induce considerable sub-lethal damage in large kelp species. The aggregation of herbivores as a consequence of cohabitation of numerous adult and juvenile conspecifics, co-existence of hetero-specific herbivores and microhabitat selection concentrates the grazing damage on structurally important stipe tissue and causes the loss of considerable biomass. The broken-off stipe sections enter the detrital pathway of coastal food webs and may become available for consumers inside and outside the sites of production [51]. These findings clearly underline the important role of small herbivores in coastal ecosystems and their contribution to the regulation of the biomass of large benthic primary producers.

Data accessibility. All data used in this study are publicly available through the data repository PANGAEA under https://doi.pangaea.de/10.1594/PANGAEA.911738.

Authors' contributions. L.G. conceptualized the study, collected data, carried out statistical analysis and drafted the manuscript; A.G.B.P. conceptualized the study, collected data, carried out statistical analysis and and helped draft the manuscript; M.A.D.P. participated in the design of the study, collected data and critically revised the manuscript; V.V. collected data and critically revised the manuscript; M.T. conceptualized the study, collected data and helped draft the manuscript. All authors gave final approval for publication and agreed to be accountable for the work performed therein.

Competing interests. We declare we have no competing interests.

Funding. Participation of A.G.B.P. was made possible by funding from FONDECYT 1161383. The authors acknowledge support from the Open Access Publication Funds of the Alfred Wegener Institute Helmholtz Centre for Polar and Marine Research.

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
