## [Reviewer comments · Proceedings of the Royal Society B: Biological Sciences]

Review History

RSPB-2020-0330.R0 (Original submission)

Review form: Reviewer 1

Recommendation

Major revision is needed (please make suggestions in comments)

Scientific importance: Is the manuscript an original and important contribution to its field?

Good

General interest: Is the paper of sufficient general interest?

Good

Quality of the paper: Is the overall quality of the paper suitable?

Good

Is the length of the paper justified?

Yes

Should the paper be seen by a specialist statistical reviewer?

No

Do you have any concerns about statistical analyses in this paper? If so, please specify them explicitly in your report.

Yes

It is a condition of publication that authors make their supporting data, code and materials available - either as supplementary material or hosted in an external repository. Please rate, if applicable, the supporting data on the following criteria.

Is it accessible?

Yes

Is it clear?

Yes

Is it adequate?

Yes

Do you have any ethical concerns with this paper?

No

Comments to the Author

This is a well written manuscript. It addresses an interesting topic - one that will be of interest even to those outside of the marine realm. I found the introduction to provide sufficient and informative background while the discussion appropriately contextualized the results. My main concerns are about the statistical approaches that were employed and how the results have been presented (please see detailed comments below). Nonetheless, I think that these can be addressed and once this has been done this will make a nice contribution to Proceedings B. I provide detailed comments below:

Abstract:

Line 33: It is unclear to someone who is reading this work for the first time what is meant by 'stages'. This is only explained in the main text.

Methods:

General comment: All analyses are underpinned by assumptions. At the moment the paper suggests that all data are normally distributed with homogeneous variances because of the tests that have been applied. However, the graphics provided in the results suggest that this is not the case. The authors are asked to carefully reconsider each stats assessment that they have done and be explicit in the manuscript about the nature of the data. Once the reader knows that the stats have been applied correctly they can have more faith in the results.

GLMs (multiple places in the text including lines 134 and the paragraph starting line 155): This analysis is in its most basic form underpinned by assumptions of normality and equal variance. Where these checked? Aligned to this issue - what distribution underpinned the GLMs i.e. if the data were not normal its possible to apply a different distribution but if this is done it needs to be stated in the methods. GLMs can only assess fixed factors so reference to a 'fixed factor' in line 158 should be removed.

Line 138: Why were the years combined? A better approach would be to compare among years, if no difference is found only then is it ok to combine them.

Paragraph starting line 155: The more appropriate analysis for assessing the density among rocks while accounting for stipe length would be a mixed effect model. Rock would be the fixed factor and stipe length should be incorporated as a random effect. The exact for of mixed model would depend on the distribution of the data.

Line 156-158: As the analysis being done is not explicitly spatial this wording is not appropriate. Rather 'To assess if domicile density differed among rocks within the kelp forest...'

Line 163 (and every other time a Chi-squared test is referred to): Which test was used? Unless the sample size is very big (some stats resources recommend $n=1000$) then an exact test (e.g. Fishers exact) should be used and not Pearsons. Please specify.

Line 173-174: I suggest leaving this out. The approach of contrasting the distribution of the domiciles with the vertical distribution of kelp biomass along the stipes is much better and the simple approach applied first does not provide much insight.

Line 184 '(c) Estimation of damage across the kelp forest': I dont have any concerns with what was done in this section but I am concerned that it is being described as assessing damage across the kelp forest. In fact only one species of kelp was considered and it is not clear how dominant this species is in the forest. To my mind what is being estimated is damage to *L. berteroa*.

Line 189: I like the approach of 'maximal potential loss of biomass'. Good way to estimate what cannot be measured.

Line 193: What are the implications of the findings re density of domiciles (i.e. density of domiciles differs among rocks) on how these stipes were collected? If these samples were collected from the same areas as the stipes for assessing domicile density then domicile density could be accounted for in the estimate.

Line 235: Were the assumptions associated with a match t test considered?

Results:

General comments:

Please present the full statistical results in tables (even if the tables are embedded in Supp Material). It provides the reader with a better understanding of the results, especially as the currently used method of reporting the GLM results leaves out information that is normally reported.

By presenting means and running parametric statistics there is the assertion that the data are normally distributed with homogeneous variance. However the descriptive stats provided on line 323 suggest great variance. This raises questions about the validity of the statistical approach applied.

Line 280 Figure 1: These figures provide evidence that the assumptions of a GLM could not have been met unless the distribution was altered from a Gaussian distribution. No information was provided in the methods though. Additionally - need to provide statistical evidence that years were the same before simply combining them.

Line 323: Variability should rather be reflected by SD - SEM is only appropriate when describing variability around means of multiple means.

Discussion:

line 430-431: Or could it suggest that *Bircenna* are out competed?

Line 440-441: But is *L. berteroa* evenly distributed with the forest? This wording suggests that it is. As per a previous comment I think it s more appropriate to talk about this results in relation to the kelp species being studied rather than at the level of the kelp forest.

Line 446 'indicates': This wording is a bit strong. 'suggests' would be better as you are making an inference.

Review form: Reviewer 2

Recommendation

Accept with minor revision (please list in comments)

Scientific importance: Is the manuscript an original and important contribution to its field?

Excellent

General interest: Is the paper of sufficient general interest?

Good

Quality of the paper: Is the overall quality of the paper suitable?

Excellent

Is the length of the paper justified?

Yes

Should the paper be seen by a specialist statistical reviewer?

No

Do you have any concerns about statistical analyses in this paper? If so, please specify them explicitly in your report.

No

It is a condition of publication that authors make their supporting data, code and materials available - either as supplementary material or hosted in an external repository. Please rate, if applicable, the supporting data on the following criteria.

Is it accessible?

Yes

Is it clear?

Yes

Is it adequate?

Yes

Do you have any ethical concerns with this paper?

No

Comments to the Author

This MS documents damage done to a brown seaweed by two species of burrowing amphipods, which can cause the loss of far more biomass than they consume, making them ecologically important grazers on the scale of the kelp forest. The study is well done and should be of interest to a broad range of readers. The following relatively minor points could be addressed prior to publication:

I think the photos in supplementary figures 1 and 2 are valuable for helping the reader gain a sense of the study system, and should be in the main MS.

Orchomenella aahu is suspected to form cavities in Ecklonia radiata after entering through storm-damaged or bleached meristodermal tissue (Haggitt & Babcock 2003, p. 1206). Please discuss the possibility that the Chilean amphipods behave similarly, in which case their impact may be to hasten breakage that was going to happen anyway rather than directly cause it. The authors'

implicit assumption is that the amphipods typically burrow into initially healthy tissue, and it would be helpful to communicate any relevant observations made on this by the authors while they were dissecting the burrows.

In section (c) of the Discussion and possibly the Introduction as well it would be worth mentioning the long-known disproportionate effects of grazing by sea urchins. I haven't got the following reference handy, but believe it describes how grazing on the holdfasts and basal stipes of giant kelp causes the loss of far more biomass than the urchins eat:

Leighton, D. L. 1971. Grazing activities of benthic invertebrates in southern California kelp beds. In: *The Biology of Giant Kelp Beds (Macrocystis) in California*. North, W. J. (ed.). Verlag von J. Cramer, Lehre, Germany. pp. 421-453.

Line 107: explain how amphipod domiciles were detected by the snorkeller in the field. Could the snorkeller see the entry hole mentioned in line 262? Or did they collect plants at random and later discard those without domiciles?

Line 171: "in a branching were" needs rewrite

Line 413: "The simultaneous occurrence of adults and juveniles in the same burrow reveals extended parental care in *S. lessoniophila*." I don't know the exact definition of extended parental care but surely it requires more than just adults and juveniles being found in the same place?

Line 443: "by up to more than 20" needs rewrite

Decision letter (RSPB-2020-0330.R0)

13-Mar-2020

Dear Dr Gutow:

Your manuscript has now been peer reviewed and the reviews have been assessed by an Associate Editor. The reviewers' comments (not including confidential comments to the Editor) and the comments from the Associate Editor are included at the end of this email for your reference. As you will see, the reviewers and the Editors have raised some concerns with your manuscript and we would like to invite you to revise your manuscript to address them.

Research ethics:

Use of animals and field studies:

Please submit a copy of your revised paper within three weeks. If we do not hear from you

within this time your manuscript will be rejected. If you are unable to meet this deadline please let us know as soon as possible, as we may be able to grant a short extension.

Best wishes,
Dr Sasha Dall
mailto: proceedingsb@royalsociety.org

Associate Editor
Board Member: 1

Comments to Author:

This manuscript finds that small amphipod grazers cause considerable damage to kelp plants by burrowing into their stipes. The bulk of the damage (which may account for between 24- 44% biomass loss in kelp forests) appears to be from plant breakage at the sites of the burrows rather than direct consumption. Both reviewers were very complimentary about this manuscript, and neither highlighted any serious problems with it. The issues raised by the reviewers are generally very minor, although one reviewer noted that the stats frequently needed better justification or explaining. It would appear that even if some of the statistical tests were to change, the main results and conclusions of the manuscript are unlikely to be affected. I would like the few comments by the reviewers to be carefully addressed, especially those concerning the justification of statistics.

I also read the manuscript and would like to draw attention to several additional issues which I would like to see addressed:

1) This manuscript frequently categorizes grazers as large or small: for example, amphipods are classified as small but urchins and fish are large. But in reality, grazer size must surely be a continuous trait. Given the continuous nature of the trait, where is the cut off drawn and perhaps more importantly, should a cut-off be drawn? Should urchins really be classified as small grazers? Are they not more similar in size to the amphipods than they are to fish? Authors need to justify the way organisms are categorized by size.

2) L38. In the abstract and I think in the discussion, the authors allude to the fact that the damage caused by the amphipods is disproportionate to their size. This assumption (that large grazers cause more damage) is not logical. It is well-known from terrestrial systems that animals of small size (e.g. ants/termites) have enormous ecological effects on their habitats. This is because despite their size, their biomass per m² is often enormous. I think that if the authors are going to try and argue that these grazers cause disproportionate damage, they need to do so in relation to the biomass (not the size) of these grazers. Unfortunately the authors never really present any comparative data to bring this point home. Is there anything that can be presented in this respect?

3) L113: I think that the authors should classify the different stages of infestation in the methods, not the results.

4) L271-273: This sentence does not make sense. Rewrite

5) L305: Adult females shared their burrow with one to several (apparently up to three) cohorts of juveniles indicating repeated reproduction of the females inside their burrows (Figure 2).....

While females may well have repeated bouts of reproduction in their burrows, I do not think that this is strong evidence for it. Juveniles could be generally using the burrows of adults to live in. How do we know that the juveniles are her own? It also appears as though females share their burrows with males too and there does not seem to be evidence suggesting that the burrows of males do not have juveniles in them.

6) Figure 4 is a plot of number of stipe breakages (X) against potential biomass loss (Y). The description of the plot is as follows: Stipe breakage induced a considerable maximal potential loss

of biomass. The estimated loss of biomass due to amphipod domiciles varied between 1.2 % induced by a single domicile (in a stipe with seven internode levels) and 77.0 % induced by four domiciles (in a stipe with nine internode levels) (Figure 4A)... I don't think that Fig 4A has been properly described in the text. For one, the text describes details which are not even seen in the plot (i.e. there is nothing in the plot about the number of domiciles). It's not really clear what information the authors are trying to convey using this plot. What I see from this plot is simply that a small number of breakages can cause a lot of damage. Is it not worth fitting some curves to these data? It looks like they could be logarithmic curves, suggesting that just a few breakages can cause maximal damage. If the authors are going to write about the number of domiciles and estimated biomass loss (as they have done), why not plot that graph?

7) The authors make quite a big deal about parental care in tis manuscript and how this has probably led to the large-scale damage of kelp. Evidence for parental care is presented in line 414: The simultaneous occurrence of adults and juveniles in the same burrow reveals extended parental care in *S. lessoniophila*.... However I do not find anything compelling about this evidence. Cohabiting a burrow has no implications about parental care. The authors need to be much more circumspect about these kinds of speculative assumptions

8) Similar to the last point about speculation, the authors write on line 426 that "During the early domicile stages the amphipod assemblages were numerically dominated by *S. lessoniophila*, suggesting that the presence of *S. lessoniophila* inhibits the presence of *Bircenna* sp. inside the shared burrows"....I do not see that the two follow from one another. It could be completely flipped around so that perhaps the old burrows of *S. lessoniophila* facilitate colonization by *Bircenna*. Hence *Bircenna* is only found in numbers in older stipes.

9) More speculation, the authors write on line 446 that "The clumped distribution of domiciles indicates that the stipe-burrowing amphipods rarely migrate beyond local kelp aggregations..." Many species have clumped aggregations and there can be many reasons for clumping which do not necessarily include anything about migration. It could simply be mating aggregations, or they may be aggregating on poorly defended kelps. Who knows. I do not see a need for the authors to give a reason for the aggregations because they really have not studied these. They just need to say that uneven distributions and aggregations of amphipods have enormous consequences

10) The authors write on Line 491 that "The missing biomass of single damaged stipes of *L. berteroa* varied between 1 % and 77 %, depending on the number of breakages per stipe and the positioning of the breakages along the stipes." However I did not take this away from the data in the paper! For example, Fig 4A suggests that one or 2 breakages are likely to cause about as much damage as 20 breakages! Yes, I agree that the damage can be highly variable, but I could not see any good statistics about biomass loss depending on the number of breakages and the point of breakage. Why not do away with the speculation and model biomass loss using number of breakages, position of breakage and perhaps even number of burrows as explanatory variables?

11) L512: No evidence for parental care

12) L515: "The loss of kelp biomass due to grazer-induced stipe breakage clearly exceeds the consumption rates of small herbivores." I agree that this is almost certainly true, but the authors present no data on consumption rates to back this up.

Reviewer(s)' Comments to Author:

Referee: 1

Comments to the Author(s)

This is a well written manuscript. It addresses an interesting topic - one that will be of interest even to those outside of the marine realm. I found the introduction to provide sufficient and informative background while the discussion appropriately contextualized the results. My main concerns are about the statistical approaches that were employed and how the results have been presented (please see detailed comments below). Nonetheless, I think that these can be addressed

and once this has been done this will make a nice contribution to Proceedings B. I provide detailed comments below:

Abstract:

Line 33: It is unclear to someone who is reading this work for the first time what is meant by 'stages'. This is only explained in the main text.

Methods:

General comment: All analyses are underpinned by assumptions. At the moment the paper suggests that all data are normally distributed with homogeneous variances because of the tests that have been applied. However, the graphics provided in the results suggest that this is not the case. The authors are asked to carefully reconsider each stats assessment that they have done and be explicit in the manuscript about the nature of the data. Once the reader knows that the stats have been applied correctly they can have more faith in the results.

GLMs (multiple places in the text including lines 134 and the paragraph starting line 155): This analysis is in its most basic form underpinned by assumptions of normality and equal variance. Where these checked? Aligned to this issue - what distribution underpinned the GLMs i.e. if the data were not normal its possible to apply a different distribution but if this is done it needs to be stated in the methods. GLMs can only assess fixed factors so reference to a 'fixed factor' in line 158 should be removed.

Line 138: Why were the years combined? A better approach would be to compare among years, if no difference is found only then is it ok to combine them.

Paragraph starting line 155: The more appropriate analysis for assessing the density among rocks while accounting for stipe length would be a mixed effect model. Rock would be the fixed factor and stipe length should be incorporated as a random effect. The exact for of mixed model would depend on the distribution of the data.

Line 156-158: As the analysis being done is not explicitly spatial this wording is not appropriate. Rather 'To assess if domicile density differed among rocks within the kelp forest...'

Line 163 (and every other time a Chi-squared test is referred to): Which test was used? Unless the sample size is very big (some stats resources recommend $n=1000$) then an exact test (e.g. Fishers exact) should be used and not Pearsons. Please specify.

Line 173-174: I suggest leaving this out. The approach of contrasting the distribution of the domiciles with the vertical distribution of kelp biomass along the stipes is much better and the simple approach applied first does not provide much insight.

Line 184 '(c) Estimation of damage across the kelp forest': I dont have any concerns with what was done in this section but I am concerned that it is being described as assessing damage across the kelp forest. In fact only one species of kelp was considered and it is not clear how dominant this species is in the forest. To my mind what is being estimated is damage to *L. berteroa*.

Line 189: I like the approach of 'maximal potential loss of biomass'. Good way to estimate what cannot be measured.

Line 193: What are the implications of the findings re density of domiciles (i.e. density of domiciles differs among rocks) on how these stipes were collected? If these samples were collected from the same areas as the stipes for assessing domicile density then domicile density could be accounted for in the estimate.

Line 235: Were the assumptions associated with a match t test considered?

Results:

General comments:

Please present the full statistical results in tables (even if the tables are embedded in Supp Material). It provides the reader with a better understanding of the results, especially as the currently used method of reporting the GLM results leaves out information that is normally reported.

By presenting means and running parametric statistics there is the assertion that the data are normally distributed with homogeneous variance. However the descriptive stats provided on line 323 suggest great variance. This raises questions about the validity of the statistical approach applied.

Line 280 Figure 1: These figures provide evidence that the assumptions of a GLM could not have been met unless the distribution was altered from a Gaussian distribution. No information was provided in the methods though. Additionally - need to provide statistical evidence that years were the same before simply combining them.

Line 323: Variability should rather be reflected by SD - SEM is only appropriate when describing variability around means of multiple means.

Discussion:

line 430-431: Or could it suggest that *Bircenna* are out competed?

Line 440-441: But is *L. berteroa* evenly distributed with the forest? This wording suggests that it is. As per a previous comment I think it is more appropriate to talk about this results in relation to the kelp species being studied rather than at the level of the kelp forest.

Line 446 'indicates': This wording is a bit strong. 'suggests' would be better as you are making an inference.

Referee: 2

Comments to the Author(s)

This MS documents damage done to a brown seaweed by two species of burrowing amphipods, which can cause the loss of far more biomass than they consume, making them ecologically important grazers on the scale of the kelp forest. The study is well done and should be of interest to a broad range of readers. The following relatively minor points could be addressed prior to publication:

I think the photos in supplementary figures 1 and 2 are valuable for helping the reader gain a sense of the study system, and should be in the main MS.

Orchomenella aahu is suspected to form cavities in *Ecklonia radiata* after entering through storm-damaged or bleached meristodermal tissue (Haggitt & Babcock 2003, p. 1206). Please discuss the possibility that the Chilean amphipods behave similarly, in which case their impact may be to hasten breakage that was going to happen anyway rather than directly cause it. The authors' implicit assumption is that the amphipods typically burrow into initially healthy tissue, and it would be helpful to communicate any relevant observations made on this by the authors while they were dissecting the burrows.

In section (c) of the Discussion and possibly the Introduction as well it would be worth mentioning the long-known disproportionate effects of grazing by sea urchins. I haven't got the following reference handy, but believe it describes how grazing on the holdfasts and basal stipes of giant kelp causes the loss of far more biomass than the urchins eat:

Leighton, D. L. 1971. Grazing activities of benthic invertebrates in southern California kelp beds. In: *The Biology of Giant Kelp Beds (Macrocystis) in California*. North, W. J. (ed.). Verlag von J. Cramer, Lehre, Germany. pp. 421-453.

Line 107: explain how amphipod domiciles were detected by the snorkeller in the field. Could the snorkeller see the entry hole mentioned in line 262? Or did they collect plants at random and later discard those without domiciles?

Line 171: "in a branching were" needs rewrite

Line 413: "The simultaneous occurrence of adults and juveniles in the same burrow reveals extended parental care in *S. lessoniophila*." I don't know the exact definition of extended parental care but surely it requires more than just adults and juveniles being found in the same place?

Line 443: "by up to more than 20" needs rewrite

Author's Response to Decision Letter for (RSPB-2020-0330.R0)

See Appendix A.

Decision letter (RSPB-2020-0330.R1)

03-Apr-2020

Dear Dr Gutow

I am pleased to inform you that your manuscript entitled "Small burrowing amphipods cause major damage in a large kelp" has been accepted for publication in *Proceedings B*.

Open Access

Paper charges

Sincerely,

Dr Sasha Dall

Associate Editor:

Comments to Author:

Dear Authors

Thank you for going to such great lengths to address all of these comments so fully. I think you have done an excellent job.

Yours sincerely

Bruce

Appendix A

Responses to reviewer comments

Associate Editor

Comment 1: This manuscript finds that small amphipod grazers cause considerable damage to kelp plants by burrowing into their stipes. The bulk of the damage (which may account for between 24 - 44% biomass loss in kelp forests) appears to be from plant breakage at the sites of the burrows rather than direct consumption. Both reviewers were very complimentary about this manuscript, and neither highlighted any serious problems with it. The issues raised by the reviewers are generally very minor, although one reviewer noted that the stats frequently needed better justification or explaining. It would appear that even if some of the statistical tests were to change, the main results and conclusions of the manuscript are unlikely to be affected. I would like the few comments by the reviewers to be carefully addressed, especially those concerning the justification of statistics.

Response: The authors appreciate the work by the Editor and the two Reviewers in evaluating our manuscript. The valuable comments, recommendations and suggestions greatly helped improving our manuscript. We have carefully addressed every single comment in our below detailed responses.

Comment 2: This manuscript frequently categorizes grazers as large or small: for example, amphipods are classified as small but urchins and fish are large. But in reality, grazer size must surely be a continuous trait. Given the continuous nature of the trait, where is the cut off drawn and perhaps more importantly, should a cut-off be drawn? Should urchins really be classified as small grazers? Are they not more similar in size to the amphipods than they are to fish? Authors need to justify the way organisms are categorized by size.

Response: It is, of course, correct that body size is a continuous trait and that it is difficult to define a clear threshold size between 'small' and 'large' herbivores. Therefore, small and large herbivores have been distinguished rather functionally. Large herbivores use seaweeds primarily as food whereas small herbivores are small enough to use seaweeds not only as food but also as (protective) habitat. Moreover, many small herbivores often inhabit tubes or domiciles on their host plants. Accordingly, their mobility, relative to the size of their food plant, is clearly limited as compared to larger herbivores. The stationary life-style of many small herbivores also has functional implications. Small herbivores have been shown to respond fundamentally differently to plant chemical defenses than do large herbivores. While large herbivores, such as fish and sea urchins, are deterred by secondary plant metabolites, small herbivores may preferentially feed on chemically defended seaweeds. The small consumers sequester the defense chemicals of the seaweed, thereby becoming less palatable to their predators, such as omnivorous fishes that feed on seaweeds and small herbivores.

Based on the above characteristics, Hay et al. (1987) established the term 'mesograzers' for small 'insect-like' herbivores, comprising mostly peracarid crustaceans but also gastropods and few polychaete species. A maximum body size of about 2 cm has been suggested for mesograzers in the literature. Meanwhile, the term 'mesograzers' is well established in the scientific literature. We decided not to provide a detailed derivation of this term and the distinction of small and large herbivores in our manuscript. Instead, we cite Hay et al. (1987) when we refer to small herbivores and mesograzers for the first time in our manuscript. In lines 49-53 it now reads: "Marine macrophytes also host diverse assemblages of small herbivores,

collectively known as mesograzers [sensu 5], which use seaweeds as food and habitat (including peracarid crustaceans and gastropods) [6-8]. Mesograzers often occur in high abundances on macroalgae and seagrasses [9], but their effects on the fitness of individual macrophytes and on the fate of benthic primary production remain difficult to predict.”

The publication by Hay et al. (1987) was added to the reference list in lines 511-512:

[5] Hay ME, Duffy JE, Pfister CA, Fenical W (1987) Chemical defense against different marine herbivores: Are amphipods insect equivalents? Ecology 68:1567–1580

Comment 3: L38. In the abstract and I think in the discussion, the authors allude to the fact that the damage caused by the amphipods is disproportionate to their size. This assumption (that large grazers cause more damage) is not logical. It is well-known from terrestrial systems that animals of small size (e.g. ants/termites) have enormous ecological effects on their habitats. This is because despite their size, their biomass per m² is often enormous. I think that if the authors are going to try and argue that these grazers cause disproportionate damage, they need to do so in relation to the biomass (not the size) of these grazers. Unfortunately the authors never really present any comparative data to bring this point home. Is there anything that can be presented in this respect?

Response: The Editor is, of course, right that damage to macrophytes should be best related to the biomass of the consumer. Unfortunately, we do not have any quantitative information on the biomass of the amphipods on *Lessonia berteroana* at Playa Blanca. Moreover, no comparable data are available on the biomass of the mesograzers and large herbivores, such as fish and sea urchins. In the revised manuscript, we added information from previous studies that at natural densities mesograzers are unable to limit the growth of macrophytes despite their considerable per capita feeding rates. Additionally, we describe that mesograzers are efficiently controlled by predators and that they can have considerable influence on macrophyte biomass if they are released from predator control. In lines 54-60 it now reads: “In many systems, the abundance of mesograzers is strongly controlled by predators, such as fish and shrimps [10,11]. Accordingly, at natural densities mesograzers are unable to limit the growth of large seaweed species [12]. Generally, the impacts of mesograzers on macrophytes are only evident when released from predator control. As a consequence of exceptional environmental conditions, or in mesocosm experiments, mesograzers can substantially increase in abundance and have strong impacts on the performance and biomass of large macrophytes [13,14].”

Additionally, we used information on body mass and individual consumption rates of grazers from a closely related amphipod species in order to provide a quantitative estimate of the amount of biomass consumed, which was then contrasted with the damage induced by the burrowing amphipods. In lines 465-471 of the Discussion it reads: “For an average stipe biomass of about 500 g (as estimated from the reconstructed stipes), the median estimated biomass loss per stipe (36.5 %) would amount to about 180 g. Adopting the body mass (0.1 g) and the daily consumption rate (about 50 % of the body mass) of adult individuals of the much larger *S. femorata* [45], average numbers of 15 amphipods per domicile and 3.2 domiciles per stipe would result in a daily consumption of about 2 g kelp biomass per stipe. At that rate, it would take about three months to consume the biomass, which is lost by grazer-induced breakage of a single stipe.”

Nevertheless, we largely avoided the use of the term “disproportionate damage” (also in the manuscript title) as the correct use of this term would require an explicit quantitative relation

of the damage to grazer biomass. The manuscript title was changed accordingly to “Small burrowing amphipods cause major damage in a large kelp”

Comment 4: L113: I think that the authors should classify the different stages of infestation in the methods, not the results.

Response: According to the Editor’s recommendation, the classification of the different stages of infestation was moved from the Results section to the Material and Methods section where it now reads in lines 113-121: “Each stipe section was photographed to document the external morphology of the domicile, which was then used to distinguish four stages of infestation; these stages likely display a temporal sequence following initial infestation (supplementary material Figure S1). In Stage 1, the domicile was apparent by a small hole formed by the initial entrance to the burrow. In Stage 2, the area of the domicile was swollen, occasionally with conspicuous deformations of the infested stipe section. The opening of the domicile was more irregular and expanded. Stage 3 was an advanced stage of tissue disintegration with gaping stretches of the stipe. In Stage 4, further disintegration of the tissue had resulted in breakage of the stipe and loss of the distal stipe sections.”

Comment 5: L271-273: This sentence does not make sense. Rewrite

Response: The sentence was rewritten. The passage now reads in lines 271-274: “The number of burrows varied significantly between domiciles of different stages of infestation (Figure 1A – for the results of the GLM see supplementary material Table S2). It was lowest in the early stages but increased substantially in domiciles of Stage 3.”

Comment 6: L305: Adult females shared their burrow with one to several (apparently up to three) cohorts of juveniles indicating repeated reproduction of the females inside their burrows (Figure 2)..... While females may well have repeated bouts of reproduction in their burrows, I do not think that this is strong evidence for it. Juveniles could be generally using the burrows of adults to live in. How do we know that the juveniles are her own? It also appears as though females share their burrows with males too and there does not seem to be evidence suggesting that the burrows of males do not have juveniles in them.

Response: The occurrence of distinct cohorts of similar-sized juveniles inside the burrow provides strong evidence for extended parental care. There are numerous examples of sub-social aggregations where individuals of all sizes aggregate, which clearly are results of gregarious behavior and which are not consequence of extended parental care (Thiel 2011). However, specific groups of one or few large adults plus one or more well-defined cohorts of juveniles are typically the consequence of prolonged cohabitation of juveniles in the burrow of their parent(s), which is referred to as ‘extended parental care’ guided by the following definition: “Any form of ‘care’ by a parent towards its offspring that has a cost to the parent, which can be in the form of space restrictions, or limited energy available for future broods. This does not have to be ‘care’ in the sense of ‘anthropogenic’ care, where adults would actively feed and nurture small offspring, but can be simple tolerance of offspring within their own dwelling (where juveniles may or may not feed on parental food resources).”

There are extensive accounts and numerous examples of extended parental care, where parents cohabit with distinct cohorts of offspring in their dwellings. We would like to invite the Editor and the Reviewers to explore the following reviews on extended parental care in crustaceans:

Thiel M (2003) Extended parental care in crustaceans—an update. *Revista Chilena de Historia Natural* 76, 205-218

Thiel, M., 2007. Social behaviour of parent–offspring groups in crustaceans. In: JE Duffy, M Thiel (eds.) *Evolutionary Ecology of Social and Sexual Systems: Crustaceans as Model Organisms*. Oxford University Press, New York, pp. 294-318

Thiel, M., 2011. The evolution of sociality: peracarid crustaceans as model organisms. In: A Asakura (ed.) *New Frontiers in Crustacean Biology*. Brill, The Netherlands, pp. 285-297

In addition to these reviews, there are many other papers on particular species that show clear evidence for well-defined cohorts of same-sized juveniles (from one brood) that have a very high likelihood of being from the same parents. We have no alternative explanation of why entire cohorts of same-sized juveniles would cohabit in the burrow of one or two large adults. Therefore, we would like to stick to extended parental care as a possible explanation for the aggregation of conspecific grazers within the burrows. In lines 389-395, we elaborated the evidence for extended parental care in *Sunamphitoe lessoniophila*: “The domiciles remain in Stage 2 for extended periods of time, which permits aggregation of numerous juvenile *Sunamphitoe lessoniophila* within the same burrow. The distinct groups of similar-sized juveniles most likely represent cohorts of juveniles from successive broods. The cohabitation of adult females (occasionally accompanied by adult males) and one to several distinct cohorts of juveniles in the same burrow provides evidence for extended parental care in *S. lessoniophila* [33]. Extended parental care is common among burrowing crustaceans and facilitates juvenile survival [34].”

Subsequently, however, we lowered our emphasis on extended parental care but simply argue that the aggregation of grazers within burrows concentrates the feeding activity on valuable tissues. For example, in the Discussion it now reads in lines 397-398: “In *S. lessoniophila*, the accumulation of conspecifics promotes the local concentration of herbivore grazing activity.” Similarly, in the Abstract it reads in lines 33-35: “Aggregations of grazers within burrows and microhabitat preference of the amphipods results in localized feeding, leading to stipe breakage and loss of substantial algal biomass.”

Comment 7: Figure 4 is a plot of number of stipe breakages (X) against potential biomass loss (Y). The description of the plot is as follows: Stipe breakage induced a considerable maximal potential loss of biomass. The estimated loss of biomass due to amphipod domiciles varied between 1.2 % induced by a single domicile (in a stipe with seven internode levels) and 77.0 % induced by four domiciles (in a stipe with nine internode levels) (Figure 4A)... I don't think that Fig 4A has been properly described in the text. For one, the text describes details which are not even seen in the plot (i.e. there is nothing in the plot about the number of domiciles). It's not really clear what information the authors are trying to convey using this plot. What I see from this plot is simply that a small number of breakages can cause a lot of damage. Is it not worth fitting some curves to these data? It looks like they could be logarithmic curves, suggesting that just a few breakages can cause maximal damage. If the authors are going to write about the number of domiciles and estimated biomass loss (as they have done), why not plot that graph?

Response: Thank you! The text was, in fact, erroneously describing the loss of biomass in relation to the number of domiciles although the figure displayed the loss of biomass in relation to the number of stipe breakages. The text was corrected accordingly so that it now reads in lines 337-342: “Stipe breakage induced a considerable maximal potential loss of biomass. The estimated loss of biomass due to amphipod domiciles varied between 1.2 % induced by a single breakage (in a stipe with seven internode levels) and 77.0 % induced by four breakages (in a stipe with nine internode levels) (median: 36.5 % – Figure 4A). Even the estimated loss of biomass induced by a single breakage varied substantially. The highest estimated loss of biomass induced by a single breakage (in a stipe with nine internode levels) was 27.9 %.”

Additionally, we followed the recommendation of the editor and fitted a non-linear regression to the data. A sigmoid model fitted the data best. The steep slope of the non-linear regression curve visualizes that already a small number of breakages can induce substantial loss of biomass. Figure 4A was modified accordingly. In lines 352-357 we added the following description: “The sigmoid regression model explained 69 % of the variation in the estimated loss of biomass for both the breakages induced by amphipod domiciles and for all breakages. 50 % of the maximal potential loss of biomass, as estimated from the sigmoid function, was reached at 1.66 breakages induced by amphipod domiciles, and at 1.77 breakages when all breakages were considered, demonstrating that already a small number of breakages can cause substantial loss of biomass.”

In the Material and Methods section, the additional regression analysis was introduced in lines 252-260 where it reads: “The loss of biomass due to breakages induced by amphipod domiciles and due to all breakages was visualized for different numbers of breakages using the following sigmoid regression model:

$$f(x) = a + \frac{(b-a)}{1 + \exp\left(\frac{c-x}{d}\right)} \quad (\text{Equation 2})$$

with a being the minimum biomass loss of stipes with no breakage, which was set equal to zero. b is the projected maximum loss of biomass, c denotes the halfway loss of biomass between a and b , and d is the slope of the curve.”

Comment 8: The authors make quite a big deal about parental care in this manuscript and how this has probably led to the large-scale damage of kelp. Evidence for parental care is presented in line 414: The simultaneous occurrence of adults and juveniles in the same burrow reveals extended parental care in *S. lessoniophila*.... However, I do not find anything compelling about this evidence. Cohabiting a burrow has no implications about parental care. The authors need to be much more circumspect about these kinds of speculative assumptions

Response: Please, see our response to Comment 6, which addresses the same issue.

Comment 9: Similar to the last point about speculation, the authors write on line 426 that “During the early domicile stages the amphipod assemblages were numerically dominated by *S. lessoniophila*, suggesting that the presence of *S. lessoniophila* inhibits the presence of *Bircenna* sp. inside the shared burrows”....I do not see that the two follow from one another.

It could be completely flipped around so that perhaps the old burrows of *S. lessoniophila* facilitate colonization by *Bircenna*. Hence *Bircenna* is only found in numbers in older stipes.

Response: We do agree with the Editor that the shifted numerical dominance of the two amphipod species inside burrows of different developmental stage can be interpreted in two opposite ways. We provide both interpretations in our manuscript in lines 404-410 where it reads: “During the early domicile stages the amphipod assemblages were numerically dominated by *S. lessoniophila*, suggesting that the presence of *S. lessoniophila* inhibits the presence of *Bircenna* sp. inside the shared burrows. When the large assemblages of *S. lessoniophila* left the burrows, the numbers of *Bircenna* sp. increased in older domiciles, which were in advanced stages of tissue degradation, suggesting that previous grazing by *S. lessoniophila* facilitates the accessibility of the algal tissue for *Bircenna* sp. [see also 37].”

Please, see also our below response to Comment 31.

Comment 10: More speculation, the authors write on line 446 that “The clumped distribution of domiciles indicates that the stipe-burrowing amphipods rarely migrate beyond local kelp aggregations...” Many species have clumped aggregations and there can be many reasons for clumping which do not necessarily include anything about migration. It could simply be mating aggregations, or they may be aggregating on poorly defended kelps. Who knows. I do not see a need for the authors to give a reason for the aggregations because they really have not studied these. They just need to say that uneven distributions and aggregations of amphipods have enormous consequences

Response: We agree that clumped distributions could arise from factors apart from limited migration, and have removed those comments about the migration behavior of the amphipods. In lines 416-424, the modified section now reads: “The effects of herbivory on plant populations and communities depend on the distribution of grazers at different spatial scales [38]. The distribution of a species is shaped by its movement behavior relative to the structuring of the landscape, i.e. the distribution and availability of habitat [39]. The amphipod domiciles were not evenly distributed among the local kelp aggregations, which are isolated from each other at Playa Blanca by variable stretches of unsuitable habitat. Some sites within the kelp forest were almost completely free of amphipod domiciles whereas at other sites, all thalli were infested. The local aggregation of herbivores results in an uneven distribution of grazing damage within plant populations [15,19] and can have implications for the structure of seaweed assemblages [40].”

Comment 11: The authors write on Line 491 that “The missing biomass of single damaged stipes of *L. berteroa* varied between 1 % and 77 %, depending on the number of breakages per stipe and the positioning of the breakages along the stipes.” However, I did not take this away from the data in the paper! For example, Fig 4A suggests that one or 2 breakages are likely to cause about as much damage as 20 breakages! Yes, I agree that the damage can be highly variable, but I could not see any good statistics about biomass loss depending on the number of breakages and the point of breakage. Why not do away with the speculation and model biomass loss using number of breakages, position of breakage and perhaps even number of burrows as explanatory variables?

Response: According to Comment 7, we added a non-linear regression analysis to visualize how the amount of damage varies with the number of breakages.

For the stipes that were used to calculate the biomass loss, we did not record the number and position of domiciles that had not yet induced breakages. Therefore, we are unable to relate the biomass loss to the number of domiciles.

Moreover, the damage cannot be related to the position of the breakages. For each stipe, we estimated the total damage resulting from all breakages. The great majority of damaged stipes had more than just one breakage so that we cannot assign a single position of a breakage to the calculated overall biomass loss. However, even though we are not able to quantify a relation between damage and the position of the breakage it is obvious that a breakage in a lower stipe section is more likely to cause a greater loss in biomass than a breakage in a more apical stipe section.

Comment 12: L512: No evidence for parental care

Response: Please, see our response to Comment 6, which addresses the same issue.

Comment 13: L515: “The loss of kelp biomass due to grazer-induced stipe breakage clearly exceeds the consumption rates of small herbivores.” I agree that this is almost certainly true, but the authors present no data on consumption rates to back this up.

Response: The Editor is right: we do not present any data on the feeding rates of the grazers on *L. berteroana* that may substantiate this statement. Please, see also our response to Comment 3, which addresses the same issue. We used quantitative information on body mass and feeding rates of amphipods of another amphipod species from the literature to estimate the loss of biomass caused by amphipod grazing. In lines 465-471 it reads: “For an average stipe biomass of about 500 g (as estimated from the reconstructed stipes), the median estimated biomass loss per stipe (36.5 %) would amount to about 180 g. Adopting the body mass (0.1 g) and the daily consumption rate (about 50 % of the body mass) of adult individuals of the much larger *S. femorata* [45], average numbers of 15 amphipods per domicile and 3.2 domiciles per stipe would result in a daily consumption of ~2 g kelp biomass per stipe. At that rate, it would take about three months to consume the biomass, which is lost by grazer-induced breakage of a single stipe.”

Reviewer 1

Comment 14: This is a well written manuscript. It addresses an interesting topic - one that will be of interest even to those outside of the marine realm. I found the introduction to provide sufficient and informative background while the discussion appropriately contextualized the results. My main concerns are about the statistical approaches that were employed and how the results have been presented (please see detailed comments below). Nonetheless, I think that these can be addressed and once this has been done this will make a nice contribution to Proceedings B. I provide detailed comments below:

Response: We thank Reviewer 1 for his overall positive evaluation of our manuscript. We carefully addressed all her/his comments. Please, see our detailed responses to each comment below.

Abstract:

Comment 15: Line 33: It is unclear to someone who is reading this work for the first time what is meant by 'stages'. This is only explained in the main text.

Response: Thank you! The sentence was rewritten so that it now reads in lines 32-33: "The composition of the amphipod assemblages inside the burrows varied between the different stages of infestation of the burrows."

Methods:

Comment 16: All analyses are underpinned by assumptions. At the moment, the paper suggests that all data are normally distributed with homogeneous variances because of the tests that have been applied. However, the graphics provided in the results suggest that this is not the case. The authors are asked to carefully reconsider each stats assessment that they have done and be explicit in the manuscript about the nature of the data. Once the reader knows that the stats have been applied correctly they can have more faith in the results.

Response: When contrasting the numbers of burrows between sampling years and among stages of infestation, we use a generalized linear model with a negative binomial error distribution. Linear models that do assume normality are inappropriate for this sort of count data and we first ran a generalized linear model with a Poisson error distribution. When that still had problems with heterogeneity of variances (checked by plots of residuals versus estimated values), we changed the error family to negative binomial. To clarify our approach, we have moved the more detailed description of the GLM, error distribution used, and method of statistical inference to the first time GLMs were mentioned in the manuscript.

In lines 123-128, the text now reads: "The number of burrows in each domicile was counted and contrasted between sampling years and among stages of infestation with a generalized linear model (GLM) and negative binomial error distribution. The generalized linear model was run with the `manyglm` function in the R package `mvabund` [30], with statistical inference from parametric bootstrapping. Assumptions of the model were checked with plots of residuals versus estimated values."

Comment 17: GLMs (multiple places in the text including lines 134 and the paragraph starting line 155): This analysis is in its most basic form underpinned by assumptions of normality and equal variance. Where these checked? Aligned to this issue - what distribution underpinned the GLMs i.e. if the data were not normal it's possible to apply a different distribution but if this is done it needs to be stated in the methods. GLMs can only assess fixed factors so reference to a 'fixed factor' in line 158 should be removed.

Response: As for Comment 16, we have clarified the analytical approach with GLMs by moving our methods text to the first time GLMs were mentioned and by adding the way in which assumptions were checked.

The reviewer is correct regarding fixed factors and the reference to a 'fixed factor' was removed so that it now reads in lines 163-165: "To test if domicile density varied between the sampling sites within the kelp forest the average domicile density was compared between the ten sampling sites by a GLM."

Comment 18: Line 138: Why were the years combined? A better approach would be to compare among years, if no difference is found only then is it ok to combine them.

Response: We agree that a sound statistical comparison would well justify the combination of the data from the two sampling years. However, after careful consideration we decided to remove this section as well as the corresponding Figure S3 from the supplementary material.

The essential information is that adult females share their domiciles with one to several cohorts of juveniles. This information is delivered by Figure 2 and the corresponding text section. The size frequency distribution (the former Figure S3) and the corresponding text section did not provide any additional information.

Comment 19: Paragraph starting line 155: The more appropriate analysis for assessing the density among rocks while accounting for stipe length would be a mixed effect model. Rock would be the fixed factor and stipe length should be incorporated as a random effect. The exact form of mixed model would depend on the distribution of the data.

Response: Our understanding of mixed models is that you would use stipe as a random effect if there were multiple measures of the response variable taken from each stipe from each sampling sites (a hierarchical sampling design). With just one measure of our dependent variable (domicile density) from each stipe, we do not consider it appropriate to use stipe as a random effect in a mixed model. To account for the different sizes of the stipes, we used stipe length as an offset in the model. In lines 163-167 it reads: “To test if domicile density varied between the sampling sites within the kelp forest the average domicile density was compared between the ten sampling sites by a GLM. Stipe length was treated as an offset to account for the higher probability of a longer stipe to become colonized by amphipods than a shorter stipe.”

Comment 20: Line 156-158: As the analysis being done is not explicitly spatial this wording is not appropriate. Rather 'To assess if domicile density differed among rocks within the kelp forest...'

Response: The sentence was modified accordingly so that it now reads in lines 163-165: “To test if domicile density varied between the sampling sites within the kelp forest the average domicile density was compared between the ten different rocks by a GLM.”

Comment 21: Line 163 (and every other time a Chi-squared test is referred to): Which test was used? Unless the sample size is very big (some stats resources recommend $n=1000$) then an exact test (e.g. Fisher's exact) should be used and not Pearson's. Please specify.

Response: Fisher's exact test is used for 2 x 2-contingency tables. We are using the chi-squared statistics for goodness of fit tests, where observed counts of a single variable are contrasted to expected counts (e.g., count of domiciles in stipe sections in contrast to expectations if randomly distributed according to availability of kelp biomass). To clarify this, we have edited the text to ensure that these are interpreted as goodness of fit tests, not contingency tables with more than one categorical variable. In lines 168-169, it now reads: “For each domicile we recorded the stage of infestation and contrasted the number of domiciles among these stages with a χ^2 -goodness of fit test.” Additionally, it reads in lines 179-182: “To test whether the distribution of domiciles among the internode levels is simply a result of a stochastic encounter of the amphipods with the stipes, we contrasted the distribution of the domiciles with the vertical distribution of kelp biomass along the stipes using a χ^2 -goodness of fit test.”

Comment 22: Line 173-174: I suggest leaving this out. The approach of contrasting the distribution of the domiciles with the vertical distribution of kelp biomass along the stipes is much better and the simple approach applied first does not provide much insight.

Response: We agree with the Reviewer's recommendation and removed the test for deviation of the vertical distribution of the domiciles from an even distribution. Now, we only contrast the vertical distribution of the domiciles with the vertical distribution of kelp biomass.

Comment 23: Line 184 '(c) Estimation of damage across the kelp forest': I don't have any concerns with what was done in this section but I am concerned that it is being described as assessing damage across the kelp forest. In fact only one species of kelp was considered and it is not clear how dominant this species is in the forest. To my mind what is being estimated is damage to *L. berteriana*.

Response: Good point and we fully agree with the Reviewer. The kelp forest definitely consists of more species than just *Lessonia berteriana* so that our estimate of biomass loss to this species does not reflect the damage induced to the entire kelp forest. Accordingly, we modified the headers of the sub-sections in Material and Methods and Results, which now read: "Estimation of damage across the kelp population". Moreover, any former reference to damage to the kelp forest now refer to damage to the local kelp population. For example, in lines 474-476 it now reads: "For the entire *L. berteriana* population at Playa Blanca, the amphipods caused an estimated loss of biomass of 24-44 %, depending on the proportion of stipe breakages that were assigned to amphipod burrowing."

Comment 24: Line 189: I like the approach of 'maximal potential loss of biomass'. Good way to estimate what cannot be measured.

Response: Thank you!

Comment 25: Line 193: What are the implications of the findings re density of domiciles (i.e. density of domiciles differs among rocks) on how these stipes were collected? If these samples were collected from the same areas as the stipes for assessing domicile density then domicile density could be accounted for in the estimate.

Response: Good point! Unfortunately, the stipes for assessing the damage were collected truly randomly within the kelp forest. We did not take care to sample the stipes from the same sites/rocks as the stipes used for the analysis of domicile distribution. Neither have we precisely documented from which sites exactly the stipes were collected. Accordingly, it will not be possible to consider the within-kelp forest distribution of the domiciles in the estimation of damage. We simply assumed that our stipes were representatively collected within the kelp forest and allow, thus, for a representative estimation of the damage.

Comment 26: Line 235: Were the assumptions associated with a matched t test considered?

Response: Yes, the differences between the matched data were tested for deviation from a normal distribution using a D'Agostino-Pearson omnibus K2 test. This information was added in lines 236-239 where it now reads: "The observed biomasses of the damaged stipes and the expected biomasses of the reconstructed stipes were compared by a paired t-test after the differences between the paired values were tested for deviation from a Gaussian distribution using a D'Agostino-Pearson omnibus K2 test (N = 17, K2 = 0.17, p = 0.08)."

Additionally, in lines 248-251 it reads: "To obtain a maximum estimate of the potential loss of biomass due to all stipe breakages (including amphipod burrows) the above calculations were also made for all breakages (D'Agostino-Pearson normality test: N = 24, K2 = 2.79, p = 0.25)."

Results:

Comment 27: Please present the full statistical results in tables (even if the tables are embedded in Supp Material). It provides the reader with a better understanding of the results, especially as the currently used method of reporting the GLM results leaves out information that is normally reported.

Response: We have added the requested tables to the supplementary material. The tables summarize in detail the GLM results for (1) the number of burrows in domiciles of different stages of infestation collected in different years (supplementary material Table S2), and (2) the number of individuals of the two amphipod species collected in different years (supplementary material Table S3). We did not add a results table for the GLM comparing the number of domiciles at different sites within the kelp forest. The detailed and complete result of this analysis is given in the main text in lines 301-303 where it reads: “The number of domiciles per stipe varied significantly among sampling sites in the kelp forest of Playa Blanca (GLM: DF = 88, deviance = 87.68, $p < 0.01$ – supplementary material Figure S3).”

Comment 28: By presenting means and running parametric statistics there is the assertion that the data are normally distributed with homogeneous variance. However the descriptive stats provided on line 323 suggest great variance. This raises questions about the validity of the statistical approach applied.

Response: With the regard to parametric statistics, please, see our response to Comment 6, which addresses the same issue.

The descriptive statistics in the former line 323 was modified to account for the non-normal distribution of the data. In lines 305-306, it now reads: “The median of the number of domiciles per stipe varied between the sampling sites from 0 (range: 0-1) to 7 (range: 3-13).”

Comment 29: Line 280 Figure 1: These figures provide evidence that the assumptions of a GLM could not have been met unless the distribution was altered from a Gaussian distribution. No information was provided in the methods though. Additionally - need to provide statistical evidence that years were the same before simply combining them.

Response: With the regard to parametric statistics, please, see our response to Comment 6, which addresses the same issue.

The results of the GLM are now presented in detail in supplementary material Table S3, which shows that the number of amphipods of the two species in domiciles of different stages of infestation did not vary between the two sampling years 2011 and 2014. This table provides the statistical evidence as requested by the Reviewer.

Comment 30: Line 323: Variability should rather be reflected by SD - SEM is only appropriate when describing variability around means of multiple means.

Response: According to Comment 28, the descriptive statistics was modified to account for the non-normal distribution of the data. The variability is now described using the median and the range of the data (please, see our response to Comment 28).

In Table S1 of the supplementary material, SEM has been replaced by SD, according to the Reviewer's recommendation.

Discussion:

Comment 31: line 430-431: Or could it suggest that *Bircenna* are out competed?

Response: Yes, the Reviewer is of course right. This option was mentioned in the sentence in lines 404-406 where it reads: “During the early domicile stages the amphipod assemblages were numerically dominated by *S. lessoniophila*, suggesting that the presence of *S. lessoniophila* inhibits the presence of *Bircenna* sp. inside the shared burrows.”

Please, see also our response to Comment 9.

Comment 32: Line 440-441: But is *L. berteroana* evenly distributed with the forest? This wording suggests that it is. As per a previous comment I think it s more appropriate to talk about this results in relation to the kelp species being studied rather than at the level of the kelp forest.

Response: Yes, we agree. Accordingly, the sentence was rewritten to better illustrate the patchy distribution of *Lessonia berteroana* within the kelp forest of Playa Blanca. In lines 419-421 it now reads: “The amphipod domiciles were not evenly distributed among the local kelp aggregations, which are isolated from each other at Playa Blanca by variable stretches of unsuitable habitat.”

Comment 33: Line 446 'indicates': This wording is a bit strong. 'suggests' would be better as you are making an inference.

Response: According to the recommendation by the Editor (Comment 10), this sentence was removed from the manuscript.

Reviewer 2

Comment 34: This MS documents damage done to a brown seaweed by two species of burrowing amphipods, which can cause the loss of far more biomass than they consume, making them ecologically important grazers on the scale of the kelp forest. The study is well done and should be of interest to a broad range of readers. The following relatively minor points could be addressed prior to publication:

Response: We very much appreciate the helpful comments by Reviewer 2. Below, we provide careful and detailed responses to each comment.

Comment 35: I think the photos in supplementary figures 1 and 2 are valuable for helping the reader gain a sense of the study system, and should be in the main MS.

Response: We agree that Figures 1 and 2 facilitate the understanding of the interaction between the amphipods and *Lessonia berteroana* as well as the rationale of this study. However, especially Figure 2 is a very large compound figure that would not allow for the inclusion of any additional figure given the strict page limitations of the journal. Accordingly, we would like to keep these figures in the supplementary material. However, in case the Editor agrees with the Reviewer and decides to relax the strict page limitations, we would be happy to follow this suggestion.

To help the reader gaining a sense for the type of damage induced by the amphipods, we added drawings of the different stages of infestation to Figure 1, which now looks as follows:

Figure 1: (A) Number of burrows within amphipod domiciles of different infestation stages (see drawings on top of the figure and supplementary material Figure S1) in stipes of *Lessonia berteriana* from Playa Blanca collected in 2011 and 2014. (B) Number of individuals of *Sunamphitoe lessoniophila* and *Bircenna* sp. in domiciles (including single burrows and conglomerates) of different infestation stages in stipes of *Lessonia berteriana*. Data from the sampling years 2011 and 2014 were combined because the pattern was similar in both years. Dot with a number above it represents an outlier at 117 individuals per domicile, which lies outside the scale of the ordinate.

Comment 36: *Orchomenella aahu* is suspected to form cavities in *Ecklonia radiata* after entering through storm-damaged or bleached meristodermal tissue (Haggitt & Babcock 2003, p. 1206). Please discuss the possibility that the Chilean amphipods behave similarly, in which case their impact may be to hasten breakage that was going to happen anyway rather than directly cause it. The authors' implicit assumption is that the amphipods typically burrow into

initially healthy tissue, and it would be helpful to communicate any relevant observations made on this by the authors while they were dissecting the burrows.

Response: Thanks for this valuable comment. Early stages of the developmental sequence of the burrows were found on undamaged stipe sections clearly demonstrating that no previous damage is needed for the amphipods to burrow into the stipes. We added this information in the Discussion in lines 380-383: “Domiciles of Stages 1 and 2 were found in healthy stipe sections demonstrating that, unlike the lysianassid amphipod *Orchomenella aahu* on the kelp *Ecklonia radiata* in New Zealand [32], the amphipod species on *L. berteroana* do not require damaged stipe tissue for the initial infestation.”

Comment 37: In section (c) of the Discussion and possibly the Introduction as well it would be worth mentioning the long-known disproportionate effects of grazing by sea urchins. I haven't got the following reference handy, but believe it describes how grazing on the holdfasts and basal stipes of giant kelp causes the loss of far more biomass than the urchins eat:

Leighton, D. L. 1971. Grazing activities of benthic invertebrates in southern California kelp beds. In: *The Biology of Giant Kelp Beds (Macrocystis) in California*. North, W. J. (ed.). Verlag von J. Cramer, Lehre, Germany. pp. 421-453.

Response: We very much appreciate that the Reviewer draws our attention to this additional reference. The fact that grazing in holdfast promotes the loss of considerable kelp biomass was already considered in our initial submission. In lines 62-65 of the revised manuscript it reads: “For example, consumption of photosynthetically active tissue by isopods affects kelp growth [15], whereas excavation of stipes and holdfasts by boring mesograzers can compromise structurally important tissues and provoke substantial biomass losses [16,17].”

We added the citation of the publication by Leighton (1971) in the Discussion as a reference in the following sentence in lines 458-460: “Grazing by herbivores on structurally important tissues can lead to disproportionate loss of plant biomass [18,47] and predicting plant damage from herbivore consumption rates alone can underestimate the effects of small herbivores on large macrophytes [48].”

The reference by Leighton (1971) was added to the list of references in lines 614-615:

[47] Leighton DL (1971) Grazing activities of benthic invertebrates in kelp beds. *Nova Hedwigia Beih* 32:421–453

Comment 38: Line 107: explain how amphipod domiciles were detected by the snorkeller in the field. Could the snorkeller see the entry hole mentioned in line 262? Or did they collect plants at random and later discard those without domiciles?

Response: The snorkeling investigator was able to see the entry holes in the stipes of *L. berteroana* allowing the infested stipe section to be cut off on the spot and transferred immediately into plastic bags in order to avoid the loss of associated amphipods. The requested information was added in lines 108-111: “The stipe sections with the domiciles were identified by a snorkeling investigator, cut off with a knife above and below the adjacent branchings of the stipe, transferred individually into plastic bags and transported in a cooler to the Universidad Católica del Norte (UCN) in Coquimbo.”

Comment 39: Line 171: "in a branching were" needs rewrite

Response: The sentence was modified so that it now reads in lines 177-178: “Domiciles positioned in a branching of the stipe were assigned to the internode level above.”

Comment 40: Line 413: "The simultaneous occurrence of adults and juveniles in the same burrow reveals extended parental care in *S. lessoniophila*." I don't know the exact definition of extended parental care but surely it requires more than just adults and juveniles being found in the same place?

Response: Please, see our response to Comment 6, which addresses the same issue.

Comment 41: Line 443: "by up to more than 20" needs rewrite

Response: The sentence was rewritten and simplified so that it now reads in lines 421-422: "Some sites within the kelp forest were almost completely free of amphipod domiciles whereas at other sites, all thalli were infested."